# Coming Home, Staying Home: Adopters’ Stories about Transitioning Their New Dog into Their Home and Family

**DOI:** 10.3390/ani14050723

**Published:** 2024-02-26

**Authors:** Eileen Thumpkin, Nancy A. Pachana, Mandy B. A. Paterson

**Affiliations:** 1School of Psychology, The University of Queensland, Brisbane, QLD 4072, Australia; n.pachana@psy.uq.edu.au; 2Royal Society for the Prevention of Cruelty to Animals Queensland, Brisbane, QLD 4076, Australia; 3The School of Veterinary Science, The University of Queensland, Gatton, QLD 4343, Australia

**Keywords:** dog adoption, post-acquisition, adopter experiences, qualitative research, transition to home, human–animal relationship

## Abstract

**Simple Summary:**

When the expectations and the reality of caring for a new dog clash, life can be challenging for adopters and their dogs. This study explored adopters’ experiences up to four years post-adoption. Many adopters in our study admitted their expectations were unrealistic and realised they were ignorant of the time and patience needed to build a positive relationship with their dog. Adopters believed building trust with the dog and learning its behavioural comfort limits in different environments was essential. Ensuring the dog’s and family members’ safety in their home was critical. Using local adopters’ stories could be a powerful tool in pre- and post-adoption support programs.

**Abstract:**

Published research estimates shelter dogs’ post-adoption returns at 7–20%, with a significant percentage of these occurring in the first month post-adoption. To better understand factors that contribute to the success or failure of long-term rehoming outcomes, this study sought to understand post-adoption challenges up to four years post-adoption, targeting dogs identified as more likely to be returned. Thirty-one adopters participated in semi-interviews. Thematic analysis of their responses yielded three themes: (1) The adoption process takes time and requires patience; (2) Building trust and learning limits are essential to lasting adoptive relationships; and (3) Human–dog relationships are idiosyncratic because they involve unique individuals. These results have potential application in programs designed to guide adopters and dogs through a successful adoption process. Access to real-life adoption stories, such as those uncovered in this study, might help new adopters develop reasonable expectations and learn from others’ experiences as they work to develop lasting relationships with their dogs.

## 1. Introduction

The adoption (re-adoption) journey for dogs and adopters can be challenging. For some dogs, finding a long-term home may take several attempts. The corollary, for many adopters, can be a success story, but for some, it can result in the difficult and traumatic decision to return their dog [1,2,3,4,5,6,7]. Accessing accurate and consistent relinquishment data across rescue organisations is problematic; however, published estimates indicate post-adoption returns of 7–20%, with a significant percentage of these dogs coming back to shelters in the first month post-adoption [4,8,9,10,11]. Notwithstanding the potential for recording inaccuracies, these estimates represent millions of dogs in shelters worldwide, once again looking for a home [12,13,14,15]. Behind these numbers, there are stories about the impact on shelter resources, on the animals’ well-being, and on the well-being of the people involved. 

How do we improve adoption outcomes for dogs? Shelter staff and researchers alike have grappled with this question for several decades. A considerable body of literature has been published on dog adoption and relinquishment. Several studies have focused on dog characteristics and human-related factors that impact adoptability or likelihood of return; factors influencing the decision to acquire a dog; and those factors that influence decisions to relinquish or rehome a dog [16,17,18]. Coe et al. [19] and Hill and Murphy [20], in their reviews of research into companion animal relinquishment, identified key themes and recurring factors influencing adoption successes and failures. Dogs’ morphology, breed makeup, age, size, behaviour, coat colour, inside or outside living, time in foster care, and health status are consistently identified as factors influencing adoptability and relinquishment [9,11,21,22,23,24,25,26,27,28]. Extensive research has also investigated owners’ reasons for relinquishment. Dog behaviour, lifestyle changes, moving, accommodation issues, a mismatch between the pet and household members, financial pressures, and lack of time are factors often identified [1,29,30,31,32,33,34]. Notably, dog behaviour issues, particularly aggression, are cited repeatedly as key influencing factors for relinquishment. However, the ‘likelihood of adoption returns is also associated with a range of owner and animal characteristics’ Powell et al. [1] (p. 1).

Much of this work aims to improve the well-being of companion animals, inform intervention and prevention programs, and assist shelters in working with the animals to improve their chances of adoption, as well as support adopters to retain, where possible, their companion animals. Several researchers have published papers on intervention and prevention approaches focusing on dog behavioural interventions at shelters, working on dog assessment tools, and providing support for adopters before and after adoption [28,35,36,37,38,39]. Findings from this research are mixed with respect to their efficacy, with some results highlighting the need for education of prospective owners, the need to tailor and target programs to suit different audiences, as well as acknowledging the challenges involved in changing human behaviour to improve welfare outcomes for pets [40,41].

The complexity and breadth of this research corpus highlight that pet acquisition, retention and relinquishment should best be viewed holistically, as a dynamic, complex, and context-specific social phenomenon [42,43,44,45]. Furthermore, as Meyer and Forkman [46] (p. 569), suggest, the ‘relationship between each owner and their dog is unique in its many nuances, and there are many variables, which could influence it and, in turn, [their] satisfaction’ with their adopted dog. The published research reflects the increasing use of mixed and qualitative methodology to gain more in-depth insights into these canine-human relationships and what might or might not influence the development of a resilient bond. 

DiGiacomo et al. [3] interviewed relinquishers to gain a detailed view of pet relinquishment, finding that for many this was a very difficult decision and experience, thus challenging a culture where relinquishers’ reasons are seen to be trivial or casual. Shore [4] also interviewed people who had recently returned an adopted dog, reporting the main reasons related to problems that arose post-adoption. This suggests that post-adoption services might help adopters tolerate the adjustment period. Marston et al. [47] (p. 358) interviewed owners of adopted dogs, investigating factors that influenced their selection of dogs and problems experienced in the first month after adoption. Their findings indicate that ‘improved matching procedures, and in-house and post-adoption training’ are strategies to improve the retention of adopted dogs. Unplanned acquisition was the focus of a paper by Holland et al. [48] (p. 13). They interviewed dog owners who had not planned to obtain a dog and reported, similar to other acquisition research, that the dog’s appearance was a key driver; however, also influential in their study was the owner’s perception of the ‘dog’s vulnerability, and the quick emotional attachment they formed with the dog’.

The canine-human relationship is multidimensional. It is continuously transacted and re-transacted on several levels within the family and in community spaces. As suggested by Cudworth [49] (p. 424), living is ‘muddied’ when you share your home with dog companions, where she expresses home as a fluid constructed place. Contemporarily, in many Western cultures, dogs are viewed as family members, best friends, and someone to come home to; thus ‘incorporating them into the most important social group of all’ the family [50] (p. 157), [51,52]. This nuanced relationship is illustrated in part by New et al. [32] (p. 198). They point out that,

‘*although many dogs are relinquished for one or more behavioural reasons, these behaviours are not unique to relinquished dogs. These behaviours are exhibited to varying degrees by dogs who remain in households*’.

This observation invites the question—What factors influence these different outcomes for similar dogs?

Our study provides unique insight by reporting the experiences of owners on their adoption journey up to four years post-adoption, at the same time ascertaining the status of that adoption. Despite a reported decline of 13.9% in the number of dogs entering RSPCA shelters over the past five years, increasingly, the dogs entering the shelters are those who take longer to rehome [53] (p. 4). Therefore, investigating more closely what happens for dogs at higher risk of being returned after adoption may offer opportunities to modify pre- and post-adoption processes and programs, and to cater more effectively to the needs of this changing population of dogs and their adopters.

### The Current Study

This study draws on results of previous quantitative research that used survival analyses to investigate risk factors associated with the readmission of dogs adopted from RSPCA Queensland shelters [11]. Of those readmitted during the two-year study (865 readmissions from a total of 6212 adoptions), just under two thirds (64%) occurred in the first two weeks post adoption, with a small number of dogs returned more than once.

Our current study aimed to explore the experiences of the people who adopted dogs that were at higher risk of being returned. For that reason, our study targeted adopters of dogs who, from our earlier study, were identified with a Hazard Ratio (HR) of ≥1.5, i.e., dogs with a 50% plus higher chance of return. As illustrated in Table 1, not all the higher-risk dogs were returned post-adoption.

Using a qualitative approach, this study intended to gain a more in-depth understanding of adopters’ experiences, and the subsequent process of making the dog part of their lives. We wanted a more holistic account of the complex and relational journey between people and their dogs [54]. As Barker and Pistrang in Camic [55] (p. 29) state, 


*The key advantage of qualitative approaches is that they can paint a vivid, subtle, and complex picture of the topic under investigation: to enable what Geertz (1973, p. 6) called “thick description.”*


Ultimately, these rich data were used to examine owners’ stories, about what happened in those first crucial weeks and subsequent years post adoption to identify what factors influenced the building of an enduring relationship with the dog or resulted in them returning or rehoming their adopted dog ([56] (p. 5), [57,58]).

Conducting qualitative research is an interpersonal activity that affects both the researcher and the participants [55,59]. Several writers in the field of qualitative research recognize the researcher as a key instrument in the research process, whose experiences and knowledge inform interpersonal activity and its recounting [60] (pp. 181–182) [58,61,62]. The researchers involved in this study work or volunteer with RSPCA Queensland and have animals from the shelter, thus bringing their personal experiences to reflect on and analyse participants’ stories.

## 2. Materials and Methods

Semi-structured interviews were used to collect qualitative data about the experiences of adopters. The interviews aimed to investigate two scenarios where possible: (i) where the dog remained in the home, and (ii) where the dog was returned to the shelter. The interview included four broad questions:Could you tell me about your decision to adopt and what things influenced you to choose [dogs name]?Can you tell me about [dog’s name] in the first couple of days once you took him/her home?What things did you do to transition your dog into the family and its new environment?In looking back at your experience with adopting your dog, is there anything the RSPCA could have done to help in supporting you through the process.

This paper focuses on the ‘transition-to-home’ data (Q3), which explores retention success. The other questions will be considered in a subsequent paper.

### 2.1. Participant Recruitment

An initial tranche of 15 dog identification (id) numbers was selected randomly from each of the four purposive outcome category groups outlined in Table 1. Related participants’ details were sourced from the ShelterBuddy© database and emails sent to this group of potential participants. The response rate to the emails and follow-up texts was extremely low, with less than five responses received.

Given this poor response, it was decided to invite participation by telephone. Acknowledging the difficulties for people to provide written consent by telephone, participant information was discussed before the interview started. Informed consent was recorded at the commencement of the interview. More dog id numbers were selected, and adopters were contacted until a meaningful set of data was collected to meet the aims of the study. In total, 130 contacts were made. Contacting people by telephone proved the most successful strategy, with just over 80% of those who answered the phone agreeing to an interview. Of those adopters contacted, 34 agreed to interview; however, three did not respond at the agreed time/s, and four later declined to participate. People who had returned the dog were less inclined to be interviewed; however, seven gave interviews.

Engaging participants proved to be challenging and took considerably longer than expected. Reluctance to reply to an unsolicited email, unknown mobile number, or a text message from an unknown or unexpected person/s or organization was perhaps exacerbated by recent significant data breeches in Australia at that time (https://www.webberinsurance.com.au/data-breaches-list#twentytwo; accessed 24 October 2023). In addition, database adopter details provided in several cases were no longer correct, i.e., phones disconnected, and emails not delivered. Interestingly, 4:00–5:30 pm proved to be the most successful time to get an answer to a phone call, and it often resulted in immediate agreement to participate in an interview.

### 2.2. Participant Demographics and Dog Profiles

Most adopters had previous experience with or had owned a dog. As illustrated in Table 2, there was a wide range of ages, with just under 65% of participants, aged between 35 and 65 years, with 77% of them living in couple or family households. Notably, although most people lived in their own homes, a small number lived in rental accommodation. More females (61.3%) than males (38.7%) were available for interviews.

The dog profiles provide a snapshot of the dogs in the stories. Most came into the shelters with little or no history. They entered as strays or owner surrenders. Prior to adoption, all dogs were behaviourally assessed, received basic training and/or underwent behaviour modifications when necessary. The majority were mixed breeds, predominately medium to large in size and weight, adolescent to mature, with a small number of dogs under a year old. Table 3 summarizes the dogs’ profiles and provides a brief outline of their background information from the shelter notes, adoption history and outcomes, where known.

### 2.3. Interview Process

One researcher, ET, conducted all the semi-structured phone interviews with participants. Interviews lasted 9 to 45 min (mean = 17 min, median = 30 min). Participants also provided demographic details. In total, 31 interviews were recorded across the four target sub-populations, with all interviewees agreeing to audio-recorded interviews. One interview record is based on handwritten notes, as the interview recording was inaudible.

As interviews were collected in late 2022 and early 2023, three to four years post-adoption, we were afforded an extended timeline to explore the longer-term outcomes of the adoptions.

### 2.4. Data Analysis

#### Thematic Analysis

Audio recordings were professionally transcribed in intelligent verbatim word document format, with participants de-identified and pseudonyms allocated to interviewees and dogs for publication purposes. Transcripts were checked against the recording for accuracy. Print copies of the transcripts with responses to the questions were made. The answers were recorded in an Excel spreadsheet with additional comments also recorded.

Data were analysed using an inductive, thematic approach, which values the ‘voices and stories’ of participants. The development of codes and themes is grounded in their data, reflecting the principles of grounded theory by ‘letting the data speak independently’ before interpretation ([63] (p. 635), [64]). Theoretically, this analytical approach offers flexibility, as codes are not researcher-driven nor prescribed by a more structured theoretical framework.

The six phases of reflexive thematic analysis (RTA) outlined by Braun and Clarke [57] guided the analysis process. Phase 1 involved repeated familiarization with the data, noting initial ideas in the Excel table. Phase 2 involved descriptive and latent coding and collation of responses to each of the interview questions. Code labels were grouped into potential themes (Phase 3), and related phrases used by interviewees were copied into a Word table with explanatory comments. Once the complete data set was re-examined, we refined the initial themes to ensure our analysis reflected participants’ experiences and knowledge. The three key themes were identified, refined, and named during the iterative write up (Phases 4–6). As the purpose of this paper was to explore the ‘transition to home’ in the finalization of the themes and the write up, the focus was predominantly on this topic.

## 3. Results

### 3.1. Participant Voices

In presenting an analysis that authenticates participants’ stories, it is important to acknowledge the willingness of people to share honestly their experiences, and the feelings these evoked for them. Their voices conveyed affection, frustration, enjoyment and in some cases grief, when speaking about their dogs.

Well, he’s been such a blessing for us truly, I just think we got so lucky with him and hopefully he would feel the same about us.(Sharon and Sheriff: fifth & successful adoption)

For those who had returned their dog, the decision was difficult. The pain and sense of failing the dog expressed by the adopters was palpable, despite the decisions being driven by the need to ensure their own and the dog’s welfare. It appears that when adopters returned the dogs soon after adoption, it was because they could not manage the dog due mainly to the size of the dog or its behaviours.

He was just a little bit bigger—it breaks my heart and it’s really hard to talk about. Yeah, I was really sad to have to take him back. I hope he’s found a lovely home because it was hard.(Patricia & Sheriff: fourth of five adoptions)

…but I took him back after the first week. I said look, I’m sorry, I just can’t handle him…(Clem & Cooper: first of two adoptions)

These adopters did not know the eventual outcome for the dogs and returning them remained a distressing memory. This sense of failing the dog was possibly more poignant as each had believed they were helping a dog in need and had adopted it from a shelter to give it a good home. Each was relieved to learn the dogs had found a long-term home.

Oh, goodness. Oh, I’m just so pleased that he found a home.(Carmel & Rusty: first of two adoption)

### 3.2. Themes

Three themes were generated in the final analysis: (1) The adoption process takes time and requires patience; (2) building trust and learning limits are essential to lasting adoptive relationships; and (3) human–dog relationships are idiosyncratic because they involve unique individuals.

These themes are not discrete as they overlap and chart the development of the unique relationship between each person and their dog, across time and place, and through the good times and the challenges. They also capture the reflections and learnings of the adopters. Overwhelmingly, the participants referred to their dog as a valued family member or close companion. Participants expressed their commitment to giving their dog a good life and a good home.

#### 3.2.1. The Adoption Process Takes Time and Requires Patience

Throughout discussions about transitioning the dog into their home, participants recalled that it took time for the dog to settle in, including interacting safely with family members, especially children, visitors, and other pets. It also involved testing how the dog would react in public spaces, including walks and visits to dog parks, beaches, and cafes. These test runs prompted owners to spend time training their dogs to become familiar with the situation or, where prudent, avoid situations that put them, the dog, or others at risk.

We let her off the lead [in the dog park], but we realised pretty early on that she just had this anxiety, obviously something happened in the past. We knew, okay, she can’t go to dog parks and stuff …(Kat and Bessy, second & final adoption)

Frequently, participants suggested it took three to six months for the dogs to trust them as owners. In turn, it was evident that owners also became more confident, trusting their knowledge about the dog’s limits and comfort levels. These observations were emphasized more often when the dogs exhibited separation anxiety, were fearful of objects such as a broom, were frightened by loud noises or became highly excited in public spaces. In one case, the dog was highly fearful of water, which was confronting for the owner. This recollection suggests that the participant realised it was not a case of teaching her to like water (his desired goal); she needed to feel confident and choose to engage with a novel environment (her goal).

I live across the road from the beach and I’m a surfer and I’d want to walk Suzie down to the beach to check the surf and she was just terrified of the ocean … I think it was only just a matter of time. So like in about six months I think she just randomly walked down to the beach with us one day and then…a few months after that, she’d like run through a puddle. …what I learnt from it was, it was just not teaching her to like the water but for her to just trust us, as owners, that she was going to be safe. As soon as she trusted us, she seemed to just get over the fear of it all.(Steve and Suzie, third and final adoption)

Participants also stressed the need for patience, freely acknowledging that this could be frustrating and challenging; however, they realized they were ignorant about or held unrealistic expectations of how long this transition could take. Several noted the possible impact of the dog’s earlier life experiences on the settling-in process, given that many of the dogs had been adopted and relinquished previously. Kirsty talked about persisting with teaching and engaging with her dog to address his habit of escaping.

I think that a big thing was for us that we were a little ignorant in how long it would take Oska to settle in. … I would say that it took him a good three to six months for him to realise home is home. …Like we definitely had an issue of him escaping and getting out and all that sort of thing, but yeah, he just needed a little bit of time to settle-in, I think. Obviously getting adopted out and sent back a couple of times sort of rocked his little confidence and that sort of thing as well. I think he was probably needing an adjustment period as well, but yeah.(Kristy and Oska, second & final adoption)

Over time, some behaviours became less challenging or ceased as the dogs became familiar with and comfortable in the various environments. Otherwise, owners accepted that some situations or activities were not safe for their dogs and were prepared to forego these. In contrast, other participants recounted that their dog settled seamlessly into its new home environment without any issues.

She settled in straight away. She sleeps inside and outside; she changes her mind.(George and Sandy, second and final adoption)

Sometimes, these stories contradicted the reported experiences of previous adopters, highlighting the very different outcomes for the same dog in a subsequent adoption; for example, Oska was returned for rough play-fighting with an existing dog and jumping on the children, while Sandy was returned for being aggressive with the adopter’s husband. In Sheriff’s case, comparing one of his four returns with his fifth long-term adoption was possible. The difference was about the adopter’s capacity, lifestyle, and home environment. The family lived in an extended family situation on acreage, which enabled them to care for and always manage a boisterous young dog with separation anxiety.

Some participants commented about not knowing the dog’s history or having information about what to expect when they took the dog home. Participants appreciated that the shelter might not have much information about their dog’s history. Several noted that the adoption personnel provided some information about the dog’s health, reasons for return, and any caveats about its behavioural needs. Only a few mentioned receiving a follow-up call or phone support soon after the adoption.

A small number of participants suggested that attending a seminar or watching a video might be a useful educative tool for potential adopters to provide insight into what an adopter might encounter. Another recounted her helpful conversation with her dog’s foster carer, as this person knew what the dog was like in a home environment. The timing and delivery methods that people preferred for accessing information, advice and support were diverse. Foster carers, RSPCA staff and volunteers were seen as sources of support.

Maybe a bit of education, … I wonder whether there should be a bit of a seminar like before you [adopt]—like you have to go and watch a video.Patricia

I think that you should be able to get someone from the RSPCA to come out, to watch what the dog’s problem is. They should have a better understanding because they’re with the animals a lot. Even some of the volunteers I found, had a lot of experience with different types of dogs. Even they helped me.Grace

#### 3.2.2. Building Trust and Learning Limits Are Essential to Lasting Adoptive Relationships

Confronting and adjusting initial expectations of owning a dog was a recurring observation. Many participants had not envisaged some of the challenges they faced once the dog came home. As discussed, caring for and successfully integrating the dog into one’s life requires time and patience. However, it also required the capability and willingness to recalibrate expectations to look after the dog and create a safe environment for the family and the dog. Kev acknowledged his unrealistic expectations and realised that the relationship with his dog was a two-way street. Ultimately, the power to make it work rested predominately with him.

But the reality for me was not having had a dog before. My expectations of him were completely unrealistic. It was a bit rough for the first couple of weeks till I came to my senses and thought, well this is a two-way street here. He tore up a few things… The first few weeks weren’t that flash for either of us until, as I say, he taught me some good life lessons about patience and caring.(Kev and Jack, second adoption and still in this home)

Similarly, other participants moderated their expectations of going on park walks, coffee outings or having a very social dog they could take anywhere. To maintain the bond with their dog, it was necessary to accept and work within safe limits.

But she never really like I don’t think she wants to hurt them [visting family’s dogs], but she wants to control them physically and she does it even to puppies, like it doesn’t matter the size of dog. So that’s just one thing we’ve had to like to come to terms with.(Steve and Suzie, third adoption and still in this home)

In contrast, for other participants the dog fitted seamlessly into their lifestyle practices.

Yeah I would take her [on outings]—yeah definitely. She was my fav little companion. She was fine with cafes and we did a lot of bush walks and things like that.(Nancy and Lollie, second adoption and still in this home adoption)

Setting boundaries and training were important strategies particularly for participants with children. This ensured safety for the family and the dog. However, only a small number of participants accessed professional training, several elected to train the dog themselves based on past experience or following advice provided at the shelter.

Yeah, it was just to make sure she was comfortable. We set boundaries. We said to the kids, that’s her bed. If she’s on there, leave her alone. If she knows her space and she gets that respect, then there’s less chance of her getting aggravated or anything.(Kat and Bessy, second adoption and still in this home adoption)

It’s a lot of training, that’s what I think it is. A lot of training and people don’t realise how much work they need to put in.(Grace and Bindi: second adoption and still in this home adoption)

Evident in several stories were the changes over time in some of the dogs’ demeanour and behaviour as they settled into the household and family. Some of these changes were managed within the home, and owners adjusted ensure the safety of dogs, other animals, and people.

But she doesn’t like men particularly, and we’ve had her, what, four years now…and she went from being silent and submissive, and now she’s quite defensive. So I now have to keep her away if we get a visitor to the house. Because she’s, yeah, she’s a bit nippy.(Susan and Willow, kept on first adoption and still in this home)

For some participants, this was not achievable. They rehomed their dog privately and did not return it to the RSPCA. These adopters saw this as shirking their responsibility for the dog. Participants expressed concern for the wellbeing of the dog when rehoming and endeavoured to find it a good home where it would be happy.

Well, it changed pretty rapidly over about a year period. But initially very good. He got along with the kids very well, he was fine with my other dog, but it was the kids that he got along with really well and took a connection with them, which is what was the problem in the end. He was a good dog in isolation, but he became protective of the kids and then wouldn’t let anyone near them and wouldn’t let our other dog near them.He eventually started attacking D, my other dog, and so we basically had to give him to a family that didn’t have any other dogs and had kids that just wanted to play with him. …He’s very happy and perfectly fine—as long they don’t get another dog.(Andrew and Sammy, kept on first adoption and rehomed 12 months later)

The story for Craig and Frank ended with the decision to euthanise just over two years after his adoption. Frank was a very large, active Dane mix who grew to 65 kg. Despite being a highly social dog, the dynamic with other animals changed. Craig was aware of this; however, when given the opportunity, Frank jumped the fence into the neighbours’ yard with dire consequences. As his owner recounted, despite his affable side, Frank became more dominant over time, and not safe around small animals.

Yep. Yeah, no, he was really, really good and very loving. He just had a little bit of a diabolical side that when it came out it had ramifications.(Craig and Frank, third adoption and kept, euthanised)

#### 3.2.3. Human–Dog Relationships Are Idiosyncratic Because They Involve Unique Individuals

Concepts of commitment and responsibility for their dog were implied if not explicitly stated by several participants. Indicating this often involves persistence and resolve to deal with the challenges presented and not to give up on the dog.

My wife, she took him for a walk within that first week we had him, and he did attack another fluffy black-and-white dog. So he became a menacing dog and I had to pay all that shit and have a cage and get inspected every year. But I wouldn’t change it for the world… We just turned him into part of the family, basically. I think that’s what made him a lot more of a teddy bear.(Shane and Patch, four stray intakes and kept on first adoption, died in 2023)

Well, oh mate, you just do what you do. I suppose a lot of people give up on what I had to go through with her and send her back to the pound or something.(Daniel and Macy, second adoption and still in this home)

Enmeshed in these stories are expressions of the enduring bond participants built with their dogs, and the meaningful role these dogs continue to hold in their owners’ lives.

I wouldn’t be without him. I had a friend whose dog was hit by a car, and he had to get pins in the leg and I asked him at the time that would have been in [the] 80s, how much did that cost you and it was $1200, and that would have been four weeks wages back then, I nearly had a coronary and gave him a serve.I look at it now and I understand, I didn’t then as I had never had that type of attachment to an animal before. That’s right, you take them on for life. I wouldn’t have persisted with him otherwise.(Kev and Jack, second adoption and still in this home)

Jack, at the time of the interview, had been through the third round of ligament surgery after the first two failed. Six months of treatment ensued, involving considerable financial costs, home care and containment, which proves Kev’s strong attachment and commitment to his dog. Jack’s adopter was unique in his ability to self-reflect and put time and effort into learning about dogs.

I started reading up a whole lot more on canines once I realised that he’s definitely an individual. Just some of the stuff out there around dog psychology. Some of the things he used to do, used to worry me.(Kev and Jack, second adoption and still in this home)

It was clear that many owners were resilient, prepared, and able to maintain lifestyle changes to provide a good life for their dog, who many consider a family member or a best friend. Throughout the interviews, participants spoke of their dog as an active participant in the human–dog relationship. As described in their stories, individual experiences and histories of the dog and adopter and the environment and community in which they lived influenced the relationship and bond. So, experiences were less fraught than others. Although each participant’s journey was unique, there were fundamental similarities across the enduring adoptions. These were attachment and commitment to the dog, the investment of time and energy into building a positive relationship within the capability and capacity of the dog and the owner, and the self-reflectiveness of owners to moderate expectations and adjust these to keep the dog in their family.

Our only problem is, and we still can’t to this day, we can’t leave him outside and all go out because he can jump every fence that is here, and he does, he jumps them and goes looking. There always either has to be someone here or he’s left in the house, not outside the house.(Sharon and Sheriff, fifth adoption and still in this home)

## 4. Discussion

In this study, we sought to gain a deeper insight into adopters’ experiences of transitioning a new dog into their home and family. Specifically, we explored how adopters built a relationship with their dogs and whether this resulted in an enduring bond and long-term adoption. Overall, our analyses highlighted that each adoption journey is unique and dynamic. It continues to evolve throughout the life of the owner-dog relationship. Adoption is only the beginning of this journey. 

The results of the analyses imply that building trust, keeping dogs and people safe, investing time, and adjusting expectations to accommodate the dog’s needs, as well as commitment, and capacity were essential factors in creating a place in one’s life for a dog. Several of these findings are consistent with published research on human–canine relationships. Trust and safety for dogs and people are recognised as influential in achieving successful long-term companion animal ownership [65,66,67]. McGreevey [68] (p. 136), succinctly states, ‘The cement of human-dog bonds is often called trust. Trust is built entirely on consistency’. 

Many participants were pragmatic and acknowledged that their initial expectations were unrealistic. However, they were able and willing to adjust and exercise patience to keep the dog. Several participants admitted it could be challenging to cope, with many acknowledging the likely influence of the dog’s previous experiences and adoption history. The mismatch between adopter expectations and the reality of companion animal ownership is frequently identified in dog adoption research as contributing to relinquishment [2,4,27,69]. In the context of these stories, participants accepted that this mismatch was their shortcoming and, therefore, something for them to moderate. 

In recounting the reality of caring for a dog, several participants believed they had the power in the relationship to create the opportunity for success, and it was ultimately their responsibility to provide the dog with a good life. These comments may reflect growing cross-disciplinary research on the construction of concepts such as commitment, responsible dog ownership, agency, and reciprocity as part of the complex Human–Animal-Relationship (HAR) phenomenon. Improving our understanding of the mechanisms that influence HAR could maximise their benefits and improve the welfare/wellbeing of humans and non-humans [63,70].

The study results suggest that the adoption journey is an ongoing and dynamic process that takes time. It includes a coming home phase and a staying home phase. Our results showed that phase one, the transition to home, can be smooth and uneventful for some adopters, finding a mutual dog–human rhythm and establishing a positive HAR. Conversely, it can take considerable time for some adopters to find that working rhythm. The second phase, staying home, involves strengthening and maintaining a life-long bond whereby adopters respond to the changes and challenges that might arise as people, dogs, lifestyles, and the environment change over time.

The stories in this study add to the understanding of the HAR. They provide a unique insight into the in-situ interplay between human capabilities and expectations and canine idiosyncrasies. These stories convey a holistic view of the messiness, the emotion, the challenges, successes and failures, and the benefits of living with a dog. Their strength lies in the diversity and honesty of adopters’ reflections and the potentially transformative power of their narratives. Importantly, they emphasize the dynamic of the relationship and paint different pictures of those first months post-adoption, which can be a time fraught with challenges; however, it is also the start of what could become a life-long bond. Reider [39] presents a compelling case for the benefits of post-adoption programs and their potential to improve adoption outcomes. She outlines the need for organisations to be clear about their goal for the program, the imperative to use adopter feedback to drive change, and the need for organisations to reassess what they see as a successful adoption and how to measure it. A number of adoption and welfare organisations in the United States and the United Kingdom endorse Reider’s assertion that post-adoption support can improve adoption outcomes [71,72].

Stories have the power to influence people, impact beliefs and teach new behaviours [73] (p. 176). Shelters could use adopters’ stories as one source of feedback from adopters to review adoption processes. In addition, shelters could investigate using local stories as part of their pre-and post-adoption support programs. For example, adopters’ stories could be captured on video, used as part of webinars, podcasts or seminars on dog adoption for potential adopters or as an on-demand resource for dog owners. Tailoring resources to meet the community’s needs ideally would involve adoption counsellors, behaviour staff, foster carers and local adopters in their design and use. 

Stories could also be used to stimulate discussion with staff on how they define and measure a successful adoption. Stories may also help work with adoption staff and volunteers to deepen their understanding of adoption processes from an adopter’s perspective. 

## 5. Strengths and Limitations

This study captured the experiences of people who had adopted a dog between three to four years prior to the interview. While this allowed us to capture the longer-term outcomes of these adoptions, results could not be generalised as it was a small purposive sample of harder-to-adopt dogs from one organisation in one state. The strength lies in the diversity and honesty of adopters’ reflections and potentially transformative power of their narratives.

When using qualitative methods, it would be optimal to interview in situ with the owner and their dog. This context provides more insight into the visual cues and the interactions between the dog and the adopter/s. There is a need to continue transdisciplinary research into developing positive human–animal relationships and shift the perspective from a predominantly humancentric view of success to a more holistic approach that includes the welfare outcomes for the dogs.

## 6. Conclusions

When the expectations and reality of caring for a dog are unmet, life can be rough and muddied. Many adopters in our study admitted their expectations were unrealistic and realised they were ignorant of the time and patience needed to build a positive relationship with their new dog. However, many invested the time to learn about their dogs (and themselves) and developed mutual trust. In the months and years that followed, most adopters and dogs settled into a liveable and more enjoyable rhythm. Preparing for the possible challenges and the work involved in adopting a dog, particularly one more at risk of being returned to the shelter, could help future adopters.

Stories could be a powerful resource available to shelters to improve the well-being outcomes for all animals in their care. As Dal Cin et al. assert, it is ‘naive’ not to employ stories, as ‘a vast body of research indicates that narratives influence their readers’, providing an opportunity to challenge expectations and change behaviour [73] (p. 176).

Using local real-life stories about dogs and their adopters as a resource for shelter staff and adopters could better prepare adopters, influence their choices and behaviours, and provide support as they transition. It could prompt potential adopters to reflect on their expectations and increase their awareness of the work needed, and the benefits gained from sharing their life with a dog.

A possible starting point is accepting that people and dogs could need support at various times throughout their relationship. Adoption is the start of this relationship journey, not the end. Research suggests proactive and ongoing access to post-adoption support can improve long-term adoption success. Further consideration of a more proactive and integrated paradigm of supporting people to acquire and care for a companion animal that continues to support people to keep their animals in their homes could improve outcomes for companion animals and their humans.

## Figures and Tables

**Table 1 animals-14-00723-t001:** Adoption outcomes for dogs with a Hazard Ratio of 1.5 or greater (2019–2020).

Outcome Categories	Number of Dogs
Returned ≤ 2 days	85
Returned > 2 to ≤7 days	81
Returned > 7 to ≤15 days	69
Not readmitted	2791
**Total**	3167

**Table 2 animals-14-00723-t002:** Demographic details of 31 participants.

Category	Number
**Age range**	
Under 25	1
25–35	6
35–45	8
45–65	12
65+	4
**Gender (sex)**	
female	19
male	12
**Family/household make-up**	
sole occupant	6
family (adults and children)	12
couple	12
Other: adult daughter and older parent	1
**Living arrangements**	
house	18
rental	5
acreage	7
Other: extended travel in a caravan	1
**Total number of interviewees**	31
Previously owned dogs	28
Not owned a dog	3

**Table 3 animals-14-00723-t003:** Dog profiles.

Name	Age (yrs/m)	Weight (kg)	Colour	Breed Mix	Background Information	Adoption History
Bessy	6 y	54	White with tan face	Bull Arab/Catahoula Leopard mix	single dog only household returned as cannot afford to keep	Returned once, kept on second adoption and still in that home
Bindy	6 y 4 m	43	Black brindle	Bullmastiff mix	surrender, owner going into an aged care facility perhaps not small breeds due to size returned, existing pet did not accept when in home environment	Returned once, adopted after trial adoption second time & still with this family
Bonnie	4 y	16	Black/tan	Pug/Shar Pei mix	abandoned by owner returned by adopter not good with existing dog friendly girl no young kids or other dogs secure fencing	Returned and kept on second adoption & still in this home
Brian	1 y 7 m	42	White/ black ears and eyes	Bull Arab mix	stray no poultry/killed chickens Second adoption failed due to harassing younger pup in household	Returned twice, kept on third adoption and still in this home
Cooper	2 y	14	Wan	Staffordshire/Daschund	no notes on file	Returned and kept on second adoption (final O not interviewed)
Candy	9 m	15	Red/white	Australian Cattle dog mix	issues with existing dog returned for resource guarding owner	Returned twice, kept on third adoption and still at this home
Daisy	2 y 5 m	21	Black brindle/ white	Boxer	sweet and social girl who would do well in a family environment training after adoption walks to burn off energy indoors to be part of family life surrendered for fighting with their other dog over food single dog home only with secure yard must meet any children as boxers can be quite boisterous	Surrender, kept on first adoption and still in this home
Dozer	11 m	21	Brindle	English Staffordshire/Rhodesian ridgeback mix	experienced dog owner, to continue training and be interactive keeping him occupied very treat motivated and willing to learn loves his walks, good on lead and loves play dates	Kept on first adoption, was privately rehomed in 2023 due to change in family circumstances
Frank	1 y 9 m	46	Black	Great Dane mix	big and strong must be fed on own returned as not good with existing	Returned twice and kept on third adoption. Remained in that home was euthanised recently, serious incident with neighbours’ animals
Jack	13 m	26	Tan/white	American Staffordshire/Ridgeback	returned as not bonding with kids 15 and 16 harassing chickens through fence, complaints euthanasia request as aggressive dog but agreed to rehome	Returned once, kept on second adoption and still in this home
Linda	2 y	21	White/ blue merle	Catahoula Leopard mix	no children under 13 needs to meet other dogs not good with existing dog	Returned once and kept on second adoption
Lola	2 y 9 m	35	White with black ears	Staghound mix	no cats or pocket pets suitable for older kids O must be used to large breeds surrendered as boisterous adopted once and returned as kids afraid of her	Returned once, kept on second adoption and still in this home
Lollie	7 y 3 m	38	Tan/white	Retriever/ Shepherd mix	no children under 16 yrs meet all family, needs to build trust returned did not get on with existing dog once home	Returned once, kept on second adoption. Remained in this home and passed away following illness in late 2021
Maizey	2 y 1 m	29	White/tan	Bull Arab/Border Collie mix	single dog home anxious, growls at meets so no off-leash parks etc. indoor dog	Kept on first adoption and still in this home
Macy	1 y 1 m	31	Brindle/white		friendly and social dog good family dog for an active home, must meet younger kids as big and strong, may be too much further training to teach manners MUST be allowed inside quite pushy and full on with other dogs but playful	Returned once, kept on second adoption and still in this home
Oska	6 m	13	Brown/ White	Shorthaired pointer/ Retriever mix	returned as play fighting too rough with owner’s staffy and too jumpy for the children deaf, follows other dogs’ body language excitable puppy	Returned once, kept on second adoption and still in this home
Patch	7 y 2 m	36	White/brindle patches	Bull Arab mix	foster parent advised P settled in nicely at his place loves being inside/outside and sneaks onto the bed given a chance toilet trained enjoys playing although can get a little mouthy when you stop playing and he wants to continue. not afraid of storms or the vacuum really enjoys human company hasn’t shown a lot of interest in the FP’s own dog	Came in as a stray four times and was claimed three. Adopted once and remained in this home for four years until he died from internal disease in 2023
Rascal	4 m	13	Brown brindle/white	Bull Arab mix	quiet pup, needs plenty of time to settle better with older kids in a calm household more confident around other calm dogs; may benefit from having a doggy friend at home for confidence experienced dog owner to take things at his pace and introduce him to new situations slowly	Adopted once and kept, found to be extremely timid and fearful of male in household, escaped after a few weeks in new home, and was not found
Rocky	1 y 8 m	33	White	Old English Sheepdog/ Wolfhound mix	big guy, make a great mate for an outgoing family not suited to someone and isn’t home often recommend 8 yrs + due to size of dog and mouthiness during restraint super dog, social and would do well having a social canine playmate adopters need to know requires regular brushing to prevent future mats	Adopted once and still in this home.
Rusty	11 m	13	Brindle	French Bulldog	seized by RSPCA Qld male breeding dog returned for excessive mounting behaviour towards resident dogs	Returned on first adoption within 24 h, kept on second adoption and still in this home
Sammie	7 m	16	Tan/white	Cattle Dog/Beagle mix	no notes	Adopted once and then privately rehomed in 2020 as over a year he became very protective of the kids against the existing dog
Sheriff	1 y 7 m	29	Brindle and white	Pointer/ American Staffordshire Terrier	Returned due to: jumping and escaping too big and strong separation anxiety destroyed equipment great while in the foster	Returned four times, kept on the fifth adoption and still in this home. (Interviewed one adopter who returned S & the one who kept S)
Sandy	8 y 8 m	34	Brown brindle/white	American Staffordshire Terrier/ Boxer mix	affectionate older girl, needs a home to enjoy her golden years can be unsure and worried with new people, especially men ART to discuss management plans and history no children, other dogs, or small animals in the home no dog parks, off lead areas etc.	Returned once as aggressive to husband, kept on second adoption and still in this home
Suzie	3 y 11 m	31	White and tan markings	Bull Arab/Shar Pei mix	very sweet, gentle girl attention barks at times during the day, but can settle herself eats, drinks, and toilets well	Surrendered due to illness as not able to afford treatment Returned twice, kept on third adoption after fostering during COVID and still in this home
Tiger	5 y 3 m	23	Tan/white	American Staffordshire Terrier mix	returned for escaping 6 times in two weeks when left alone throughout day has been nice and relaxed sitting on his bed or by the wire uses enrichment well dog reactive, has had long stint in kennels due to heath treatment	Returned once and kept on second adoption (final O not interviewed)
Walter	6 y	29	White/tan	American Bulldog mix	very calm, low key and has good manners ideal dog to welcome indoors with the family for bonding adopter with dog experience due to minor resource guarding if other dogs must be fed separately children over 8 yrs secure yard	Adopted once and kept, died from snake bite in 2022
Willow	4 y 6 m	30	Brindle/white	Great Dane mix	gentle natured and social girl good manners and easy to handle/walk, good family, or companion dog not recommend for young kids as they may be too noisy and unpredictable for her, suited to teens enjoys her humans, must be allowed inside, and included in the family, single dog only home	Adopted once, kept and still in this home
Whisky	13 m	22	Fawn/ cream	American Staffordshire Terrier mix	returned due to: digger not good with existing dog not safe with livestock and poultry	Returned twice, kept on third adoption and still in this home
Xavier	4 y 2 m	27	Blue roan	Cattle Dog/ Wolfhound mix	returned as kept escaping no fences on property and not safe with existing cats and dogs active home, allow indoors take to training likes food and toys worried by storms, keep inside during storms high prey drive no cats or small animals, including birds single dog home ONLY no dog parks	Returned twice, kept on third adoption

## Data Availability

The data collected for this study are not publicly available due to the ethical approval of participant informed consent, which assured them that all personally identifiable information would be removed before analysis and prior to publication.

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
