# Peer review of "Coming Home, Staying Home: Adopters’ Stories about Transitioning Their New Dog into Their Home and Family"

_animals, 2024, doi:10.3390/ani14050723_

Round 1
Reviewer 1 Report
Comments and Suggestions for Authors
The manuscript starts off with a useful commentary on the literature. The methodology is well set out – and the authors did not underplay the difficulties involved in obtaining participants (which I suppose tell a story in itself!)
The manuscript is also very well researched and the findings clearly presented (lines 229-274).
The conclusion is also well argued - starting at about line 592 – that “adoption journey is an ongoing and dynamic process”
The only point I would mention (and this does not detract from the manuscript) is that it would have been helpful to tease out the issues a little more in the discussion section; perhaps include some of the writers’ own input eg – lines 266-269 refer to the fact that dogs come with little or no history (although their adoption history might be relatively well known). Any ideas on how this contributes to the animal’s return to the shelter. Can this be overcome by behavioural analysis of the dog prior to adoption.
This is an important study. In Australia, jurisdictions are increasingly looking at making rehoming mandatory for some animals (dogs, cats, rats, guinea pigs) after experimentation. Studies such as this one generates knowledge that will be helpful for all stakeholders.
Author Response
Thank you for taking time to review this manuscript. Your supportive feedback is appreciated. Please find the response to your comment outlined below.
Comment: The only point I would mention (and this does not detract from the manuscript) is that it would have been helpful to tease out the issues a little more in the discussion section; perhaps include some of the writers’ own input e.g. – lines 266-269 refer to the fact that dogs come with little or no history (although their adoption history might be relatively well known).
Response: The issue of background knowledge about the dog's pre-shelter or in a home/foster environment has been included. (3.2.1, line 346-366) Any ideas on how this contributes to the animal’s return to the shelter.
All dogs are behavioural assessed prior to adoption (line 330) and that knowledge is used to make recommendations about the type of home suitable for them. Adopters who kept their dog were able to observe and learn about the dog’s limits and demonstrated or learnt to manage the transition challenges. Those who returned the dogs accepted that the dog and them were not a manageable fit. For very valid reasons, including energy level, size, care needs and responses to existing pets. I am not sure these challenges would be addressed with more behaviour analysis or intervention prior to or post adoption, in these cases. Perhaps a more targeted the pre-adoption discussion with these adopters could have elicited the potential for a less-than optimal fit. The ability of staff to tease out some of the potential ‘roadblocks’ is other touch point. (line 199)
Reviewer 2 Report
Comments and Suggestions for Authors
General comments
This manuscript seeks to address the lack of qualitative information on the reasons for successful or failed adoption of a dog. Another aim of the manuscript is to describe the first few days in the adopted dog's new home by focusing on four questions in particular:
. Could you tell me about your decision to adopt and what things influenced you to choose [dogs name]?
. Can you tell me about [dog’s name] in the first couple of days once you took him/her home?
. What things did you do to transition your dog into the family and its new environment?
. In looking back at your experience with adopting your dog is there anything the RSPCA could have done to help in supporting your through the process?
The idea is worthy of attention, especially since scientific papers often only report quantitative dimensions, but the main problem with this manuscript is that I see no relationship between the four questions in the Materials and Methods and the results. In this form, the manuscript turns out to be "only" a description of the dogs and their situation. In this form, it is not useful for the readers. The authors must relate the Materials and Methods and the results.
Minor points
Table 2 and 3 should be in Materials and Methods, not in Results. The tables illustrate the identity of human participants and the characteristic of the dogs involved. They are not results but the subjects of the study.
The ‘voices’ of the humans are, maybe, too many.
Author Response
Comment 1:This manuscript seeks to address the lack of qualitative information on the reasons for successful or failed adoption of a dog. Another aim of the manuscript is to describe the first few days in the adopted dog's new home by focusing on four questions in particular:
- Could you tell me about your decision to adopt and what things influenced you to choose [dogs name]?
- Can you tell me about [dog’s name] in the first couple of days once you took him/her home?
- What things did you do to transition your dog into the family and its new environment?
- In looking back at your experience with adopting your dog is there anything the RSPCA could have done to help in supporting your through the process?
The idea is worthy of attention, especially since scientific papers often only report quantitative dimensions, but the main problem with this manuscript is that I see no relationship between the four questions in the Materials and Methods and the results. In this form, the manuscript turns out to be "only" a description of the dogs and their situation. In this form, it is not useful for the readers. The authors must relate the Materials and Methods and the results.
Response 1: Thank you for raising this concern with respect to the lack of clarity on the relationship between the methods and results. We have included a statement in this section (Line 287 p.4) to indicate that this paper focuses primarily on (Q3), which explores retention success. All four questions are presented in the Materials and Method for completeness; however, the other questions will be considered in a subsequent paper.
The aim of this paper was to explore the ‘transition to home journey’ and as the interviews were held 2-4 years after adoption, the data on this aspect of the journey were extensive. In the finalization of the themes and the write up, the focus was predominantly on this topic. This is now noted thematic analysis 2.4.1 (Line 428 p. 10 )
The Results (Line 449 p10) section has also been revised to lessen the descriptive nature of the results section and add more interpretive content. Edits are tracked in the revised manuscript.
Comment 2: Table 2 and 3 should be in Materials and Methods, not in Results. The tables illustrate the identity of human participants and the characteristic of the dogs involved. They are not results but the subjects of the study.
Response 2: These tables have been moved as suggested.
Comment 3: The ‘voices’ of the humans are, maybe, too many.
Response 3: Agree. We have reviewed and reduced the participant quotes (raw data) throughout the paper. Edits are tracked in the revised manuscript.
Reviewer 3 Report
Comments and Suggestions for Authors
Although the online sourcing of data is always problematic, the authors established a fairly interesting data set and identified significant themes. I found myself growing impatient with the excessive presentation of raw data, which went on for pages without much analysis--i.e. multiple narrative snippets illustrated the same point. I would have liked the authors to do more of the analytic work--boring deeper into the stories, making more comparisons perhaps--or simply not presenting so much data.
In the discussion, I wanted to know more about the practical applications of the knowledge derived from this study. How can the "use" of real-life stories be implemented to good effect? Who should be "giving" adopters appropriate information and strategies" and in what form should they present them? What, specifically, can adopters derive from such stories? Is the same information appropriate to all dogs, adopters, and settings? Who decides what information and strategies are appropriate, and how to they make the assessment?
The simple summary is unhelpful, and the abstract is verbose and must be pared down. An abstract is no place for rhetorical questions. When I read an abstract, I am looking for the answer to three big questions:
What did you want to find out?
How did you go about searching for answers?
What did you discover?
Comments on the Quality of English LanguageThe English is perfectly readable, although expression is often wordy and indirect, and the authors need to be more careful in copy editing their work. Punctuation is erratic, and typos and misspellings are fairly common.
Author Response
Thank you very much for taking the time to review this manuscript. Revisions are in the attached file.

Round 2
Reviewer 2 Report
Comments and Suggestions for Authors
The manuscript has been improved and can be published.
Author Response
Thank you again for taking the time to review responses to our initial comments. Constructive feedback is always welcomed.
We note that Reviewer 2 supports the publication of the manuscript and we accept the comment that improvement is often possible. Feedback provided will be useful in future manuscript development.
Reviewer 3 Report
Comments and Suggestions for Authors
This is an interesting and useful study that supports the claim that the adoption process extends well beyond the initial meeting and adoption. This in turn suggests that shelters might improve adoption success rates by extending support services to adopters and dogs as the adoption develops. The presentation of narrative data further supports the claim that narratives can be useful in developing such services.
While the authors have responded to my questions and suggestions, I would like to see more work on streamlining the presentation of the study--getting directly to the point in presenting the data, the findings, and explaining the implications. For instance, there is no need in a journal such as this for the authors to rehearse the argument that qualitative data has value. I've taken the liberty to revise the abstract to illustrate what I mean by getting more directly to the point:
Published research estimates shelter dogs’ post-adoption returns at 7-20%, with a significant percentage of these occurring in the first month post-adoption. To better understand factors that contribute to the success or failure of long-term rehoming outcomes, this study sought to understand post-adoption challenges up to four years post-adoption, targeting dogs identified as more likely to be returned. Thirty-one adopters participated in semi-interviews. Thematic analysis of their responses yielded three themes: 1) The adoption process takes time and requires patience; 2) Building trust and learning limits are essential to lasting adoptive relationships; and 3) Human-dog relationships are idiosyncratic because they involve unique individuals. These results have potential application in programs designed to guide adopters and dogs through a successful adoption process. Access to real-life adoption stories, such as those uncovered in this study, might help new adopters develop reasonable expectations and learn from others’ experiences as they work to develop lasting relationships with their dogs.
Comments on the Quality of English LanguageThere are still too many careless errors and grammatical infelicities. More careful copy editing is in order.
Author Response
Thank you again to the Reviewer 3 for taking the time to review responses to their initial comments. Constructive feedback is always welcomed.
In response to comments from Reviewer 3, we have accepted the suggested revisions of the Abstract and the theme titles. We appreciate the time and thought involved in providing this feedback. The revised theme titles are much more succinct and clearly capture the intent of the themes.
‘Published research estimates shelter dogs’ post-adoption returns at 7-20%, with a significant percentage of these occurring in the first month post-adoption. To better understand factors that contribute to the success or failure of long-term rehoming outcomes, this study sought to understand post-adoption challenges up to four years post-adoption, targeting dogs identified as more likely to be returned. Thirty-one adopters participated in semi-interviews. Thematic analysis of their responses yielded three themes: 1) The adoption process takes time and requires patience; 2) Building trust and learning limits are essential to lasting adoptive relationships; and 3) Human-dog relationships are idiosyncratic because they involve unique individuals. These results have potential application in programs designed to guide adopters and dogs through a successful adoption process. Access to real-life adoption stories, such as those uncovered in this study, might help new adopters develop reasonable expectations and learn from others’ experiences as they work to develop lasting relationships with their dogs.’
In response to the suggestions of streamlining I have edited the Thematic Analysis section (p. 10; Lines 214-239). Other minor edits have also been made and tracked in the document to improve consistency.
While the authors have responded to my questions and suggestions, I would like to see more work on streamlining the presentation of the study--getting directly to the point in presenting the data, the findings, and explaining the implications. For instance, there is no need in a journal such as this for the authors to rehearse the argument that qualitative data has value. I've taken the liberty to revise the abstract to illustrate what I mean by getting more directly to the point: